# Physical Activity, Quality of Live and Well-Being in Individuals with Intellectual and Developmental Disability

**DOI:** 10.3390/healthcare12060654

**Published:** 2024-03-14

**Authors:** Susana Diz, Miguel Jacinto, Aldo M. Costa, Diogo Monteiro, Rui Matos, Raul Antunes

**Affiliations:** 1Department of Sport Sciences, University of Beira Interior, 6201-001 Covilhã, Portugal; sucris.diz@gmail.com (S.D.); mcosta.aldo@gmail.com (A.M.C.); 2Research Center in Sport Sciences, Health Sciences and Human Development (CIDESD), 5001-801 Vila Real, Portugal; diogo.monteiro@ipleiria.pt (D.M.); rui.matos@ipleiria.pt (R.M.); raul.antunes@ipleiria.pt (R.A.); 3School of Education and Social Sciences (ESECS), Polytechnic of Leiria, 2411-901 Leiria, Portugal

**Keywords:** physical activity, physical exercise, sport, quality of life, well-being, intellectual and developmental disability

## Abstract

The practice of physical activity, exercise and sport has many benefits for the general population, but studies on the population with intellectual and developmental disabilities (IDD) are scarce and inconclusive. The aim of this systematic review is to analyze the state of the art on the role of physical activity, exercise and sport in the quality of life and well-being of people with IDD, seeking to understand the current panorama in this area and provide answers to these questions. The research was carried out between July and October 2023 using three databases: PubMed, Web of Science and Scopus. Fifteen articles met the eligibility criteria for this study, and these were methodologically assessed using the Downs and Black scale (1998). Higher values were identified in the total quality of life score and some domains of this variable (e.g., personal development, physical well-being and emotional well-being), in satisfaction with life and in the perception of well-being in individuals with IDD who have enjoyed or are involved in physical activity, physical exercise and/or sports programs. Thus, according to the studies included in this systematic review, we can conclude that the practice of physical activity, physical exercise and sport seems to contribute to improving the quality of life and well-being of people with IDD. Despite the growing research interest in this area, there is still a notable lack of studies exploring the impact of these programs, especially sports-based programs, on quality of life and well-being variables in the population under study.

## 1. Introduction

Intellectual and developmental disability (IDD) is a developmental disorder that originates during the individual’s developmental period, up to the age of 22. It is characterised by limitations in adaptive behaviour and intellectual functioning, which are expressed in the conceptual, practical and social domains [1] with different degrees of severity, including mild, moderate, severe or profound. People with IDD tend to have difficulties in executive functioning [2] and psychomotor functioning, which affect their performance in activities of daily living and social participation [3], with mobility limitations [4] resulting from lower tonic and muscular performance, associated with sensory deficits and less precise and slower motor responses [5,6].

This population usually has a sedentary lifestyle [7], with low levels of physical activity (PA) [8,9,10], which tends to negatively affect their quality of life (QoL) [11]. Research has highlighted the positive impact of PA and physical exercise (PE) in preventing diseases such as type II diabetes, hypertension and obesity [7,12], as well as their contribution to improving physical fitness, with benefits for the individual’s general state of health, QoL and average life expectancy [13,14]. In this sense, different studies have shown that the practice of PA [7,15], PE [6,16] and sport [17,18] seems to be associated with an improvement in the perception of well-being and QoL of adults with IDD. It is also possible to verify that higher levels of functional capacity seem to contribute to higher QoL values [19].

The QoL of people with IDD is defined as a person’s perception of their position in life, their goals, standards, cultural and personal values and expectations in a variety of dimensions [20], and is influenced by personal characteristics and environmental factors that change over time [21]. Like QoL, well-being is also a complex concept which can be analyzed from two different but complementary perspectives—objective and subjective well-being [22]. Specifically, subjective well-being encompasses the cognitive (life satisfaction) and emotional (affect) evaluations a person makes about their life [23], and is considered a long-term state that comprises these two dimensions [24]. With regard to affect, its concept has two relatively independent dimensions [25,26]: positive affect, reflecting the extent to which a person feels enthusiastic, active and alert, and negative affect, reflecting the extent to which a person feels distressed and lacks feelings of pleasure.

In turn, the concept of life satisfaction is related to the satisfaction a person feels with life, in general or specifically in different domains (e.g., health, future security and relationships) [27], and is also associated with the social and economic resources they have [28,29]. This judgement depends on the comparison between current circumstances and what the subject defines as an appropriate standard [24,30].

Nowadays, maintaining or improving QoL and well-being is seen as a universal goal throughout the life of individuals with IDD, requiring the identification or development of facilitating tools and strategies [1]. Although several systematic reviews show that the practice of PA, PE or sport improves these variables in the adult and elderly population without disabilities [31,32], studies in the population with IDD are scarce and inconclusive. The study by Carmeli et al. [33] examining anxiety and QoL assessments revealed a 50% enhancement in the exercise intervention group, a 38% amelioration in the leisure activity group, and no improvement in the control group, providing indication that these variables may be related.

Thus, the aim of this systematic review is to analyze the state of the art on the role of physical activity, exercise and sport in the QoL and well-being of people with IDD, seeking to understand the current panorama in this area and provide answers to these questions.

## 2. Materials and Methods

The present systematic review was performed according to the PRISMA (Preferred Reporting Items for Systematic reviews and Meta-Analyses) protocol [34] and the methods suggested by Bento and collaborators [35]. The protocol was registered in PROSPERO, with the number CRD42023446863.

The PICOS strategy [36] was used to obtain a final sample of studies that included (i) a “P” (Patients) population with intellectual developmental disabilities, of any age, gender, ethnicity or race; (ii) an “I” (Intervention) corresponding to the sample practicing physical activity/physical exercise/sport or having been exposed to an intervention of the same nature and an assessment regarding their quality of life and/or well-being; (iii) a “C” (Comparison) corresponding to comparison between groups; (iv) an “O” (Outcome) corresponding to levels of quality of life and well-being; and (v) an “S” (Study design) corresponding to cross-sectional studies, pilot studies and intervention studies, randomized controlled trials (RCTs) and non-RCTs.

### 2.1. Information Sources and Research Strategies

The systematic search for articles was conducted between July and October 2023, in English, in three electronic databases: PubMed, Web of Science and Scopus (title, abstract and keywords).

The following indexed search descriptors were used across all databases in the following formats: (“mental retardation” or “intellectual disability” or “intellectual and developmental disability” or “Intellectual Disabilities”) AND (“quality of life” or “wellbeing” or “well-being”).

### 2.2. Eligibility Criteria

The inclusion criteria considered for the selection of studies were as follows: (i) cross-sectional studies, pilot studies and intervention studies, RCTs and non-RCTs.; (ii) no restrictions on age or gender; (iii) no limitations on ethnicity or race; (iv) no restrictions on the total number of participants; (v) no limitations regarding the duration, intensity, volume and frequency of programs; and (vi) studies that reported results on the effects of PA, PE or sports on QoL and or well-being. The primary exclusion criteria were as follows: (i) literature or narrative review articles, comments, or abstracts published in conference or congress proceedings; and (ii) studies in which QoL or well-being was not assessed.

### 2.3. Selection and Data Collection Process

The eligibility criteria were defined by all authors. The selected studies were reviewed in their entirety by two independent reviewers (SD and MJ), considering the eligibility criteria. The results obtained were compared and discussed to reach an agreement. When consensus was not possible, a third researcher was invited to collaborate (RA). Duplicate articles and all articles that did not meet the eligibility criteria were excluded. In the second phase, the full text was read, and the studies were selected for final analysis. After extraction, study information was described and structured in a table, namely authorship, year of publication, country, objectives, participants, duration/frequency, exercise/intensity, measurements, results, and methodology quality.

## 3. Results

### 3.1. Selection of Studies

The initial search, carried out in the three databases mentioned above, resulted in a total of 9178 articles. In the first phase, after removing duplicate articles, 6135 articles remained to be analyzed in the following phases. After reading the titles and abstracts, 6005 articles were excluded, and 130 potentially relevant articles were identified for the following analysis. Considering the previously defined inclusion and exclusion criteria, after reading the articles in full, a sample of 15 articles were considered for complete analysis (Figure 1).

### 3.2. Methodological Quality

The Downs and Black Scale [37] were used to assess the methodological quality of the studies. The scale consists of 27 items that characterize the different parts of an article. Each item has four possible answers: “yes” (one point), “no” (zero points), “does not apply” and “unable to determine” (zero points). The methodological quality of the studies was assessed independently by two researchers (SD and MJ). The results obtained were compared and discussed in order to reach a consensus. When this consensus was not possible, a third researcher was invited to collaborate (RA).

One of the questions (item 27) from the scale was removed because it was not applied to all the studies analyzed. In this way, the scale had a total of 26 questions. The methodological quality of the studies was between “Good” and “Poor”; however, no study was excluded due to its methodological quality. This classification is presented in Table 1.

### 3.3. Characteristics of the Studies

Table 1 presents the characteristics, results, and methodological quality of each of the studies included in the present review.

### 3.4. Origin

Of the 15 studies included in the systematic review, nine are from Europe (Portugal: [15]; Spain: [17,38,40,42,44]; United Kingdom: [18]; Ireland: [39]; Sweden: [41]), two are from Asia (Israel: [6,16]); two are from North America (USA: [45,47]), one is from Oceania (Australia: [46] and one is Eurasian (Turkey: [43]).

### 3.5. Type of Studies

Of the 15 studies analyzed and included in the present systematic review, 6 of them are cross-sectional studies and 8 of them are intervention studies.

### 3.6. Participants

Out of a total of 1026 participants, 995 are people with IDD, while the remaining participants are family members of people with IDD, teachers, technical carers, and association presidents. With regard to the age of the participants, it mostly ranges from 16 to 66 years old [15,16,17,18,38,40,41,42,44,46,47], with only two studies using samples of children and adolescents aged between 4 and 14 [43,45] and one study with participants aged between 46 and 77 [6]. Only one of the included studies made no reference to the age of the sample [39]. All but one of the studies [39] assessed the QoL and/or well-being of the sample, with eleven studies showing positive results regarding the impact of PA, PE or sport on the QoL and/or well-being of the sample in question.

### 3.7. Exercise Prescription

Regarding the well-being variable, the studies that aimed to understand the influence of PA, PE or sports programs on this variable used the following exercise prescription.

In the study by Barnet-Lopez et al. [38], the program implemented aimed to develop rhythm, Laban effort, balance, coordination, grounding and free dance. In the studies carried out by Carmeli et al. [6] and Carmeli et al. [16], the physical exercise programs developed by the authors focused mainly on balance training, flexibility and general strength training. The participants in the study by Carmeli et al. [6] performed 1 to 2 sets of 8 to 10 repetitions, with a rest of 2 to 4 min between sets. In the study by Shields et al. [46], the participants had a goal of 150 min of moderate-intensity PA a week, taking two 45 min walks a week, accompanied by mentors, with the participants being responsible for the remaining 60 min of PA a week.

Regarding QoL, the studies included in this systematic review used the following exercise prescription.

In the study by Fjellstrom et al. [41], the main focus of the exercise program was balance training, flexibility and general strength training, with different levels of progression (e.g., jumping or walking) being implemented in order to respond to the needs of each participant. In the studies carried out by Ozkan and Kale [43] and Snapp et al. [45], the intervention programs had a more playful character, with the Ozkan and Kale [43] study implementing educational games that promoted basic motor skills with progressions over time, taking into account the children’s progress. Snapp et al. [45] implemented sessions with free play and direct instructions also aimed at promoting children’s motor skills.

In the study by Pérez et al. [44], the participants completed a PE program in an aquatic context, consisting of a 15 min warm-up, with breathing exercises and crawl kicks, with two sets of three repetitions × 30 s each, with a rest of 5–10 s between repetitions and 1 min between sets. The fundamental part, which consisted of swimming techniques with a high swimming speed, was organized into three sets of two repetitions × 15 m, with a 10 s passive rest and 1 min of active rest between sets. Finally, in the study by Diz et al. [15], the study population took part in a PA program based on rhythmic exercises, muscle strength and amplitude, spatial orientation, balance, body awareness, attention and memory.

Given this population’s tendency towards a sedentary lifestyle [7], which tends to negatively affect QoL [11], with lower levels of functional mobility, essential for carrying out daily life tasks [14], participation in programs of this nature seem to have positive and significant effects on the QoL and well-being of people with IDD.

### 3.8. Evaluation Techniques

Different instruments were used to assess QoL in the studies presented, such as the Personal Outcomes Scale–Spanish Adaptation [40], the Life Experiences Checklist (LEC) [18], semi-structured interviews [42], the Quality of Life Questionnaire (QOL-Q) [47], the Nottingham Health Profile [6], the Portuguese version of the Personal Outcomes Scale (P_POS) [15], the Manchester Short Assessment of Quality of Life (MANSA) [41], the Pediatric Quality of Life Inventory (PedsQL) [43,45], short modified version of a Spanish QoL questionnaire and the World Health Organization Quality of Life-BREF (WHOQOL-BREF) [44].

The same is true for the well-being assessment instruments, which included the Human Figure Drawing Test (HFD) [38], focus group interviews, individual semi-structured telephone interviews, supplementary qualitative data extracted from four open ended questions contained in a quantitative survey [39], Harter’s Self-Perception Profile Modified [6,16], the Satisfaction with Life Scale (SWLS) [17] and the Life Satisfaction Scale [46].

## 4. Discussion

The main aim of this systematic review was to analyze the state of the art on the role of PA, PE and sport in the QoL and well-being of people with IDD, seeking to understand what has been established and to contribute to future studies in the area.

### 4.1. Quality of Life

Carbó-Carreté et al. [40] found that PA levels have an impact on the QoL of people with IDD, and the results obtained seem to confirm that PA acts as an important predictor of increased QoL (β11 = 0.703, *p* < 0.001). Also, the study carried out by Tomaszewski et al. [47], with the aim of analyzing the relationship between the number of steps taken and QoL, showed that the total QoL score, as well as the competence domain, is significantly associated with the weekly step count, having found that for every increase of 1000 steps per day, the total QoL increases by 2.56 points.

Crawford et al. [18], when comparing the impact that sport has on the psychosocial behavior and QoL of people with IDD, found that the sample in their study, composed of Special Olympics athletes, showed differences in the total QoL score (23.4 ± 3.7) and in the leisure and socialization domains when compared to the other groups (*χ*^2^(2, 101) = 10.1; *p* = 0.006), namely individuals who do not practice any type of sport (21.2 ± 3.7) or people who practice PE/sport but not through the Special Olympics (21.0 ± 3.5). Mercado and colleagues [42] also analyzed the possible benefits of dance in promoting the QoL of people with IDD and found that it provides benefits to the QoL of the population under study, particularly in the domains of personal development, physical and emotional well-being and interpersonal relationships.

In the study carried out by Carmeli et al. [6], after the end of the PE intervention program, the authors found statistically significant differences in the domains of energy (*p* = 0.001), social isolation (*p* = 0.001), physical mobility (*p* = 0.001) and the total QoL score in the experimental group. Analysis of variance found statistically significant differences between the groups in the domains of social acceptance (F = 8.79) and physical appearance (F = 3.15). Also in line with these results is the study carried out by Diz et al. [15], which analyzed the effects of a PA intervention program on the adaptive behavior, physical fitness and QoL of adults with IDD, and their results revealed significant differences in the total QoL score in the experimental group between the pre- and post-intervention moments (97.75 ± 8.46 vs. 101.750 ± 7.69, *p* = 0.01, r = 0.32).

PA also seems to promote QoL in children with IDD, and this can be seen in the studies carried out by Ozkan and Kale [43] and Sanpp et al. [45]. In the study conducted by Ozkan and Kate [43], with the aim of verifying the existence of differences in QoL between children who practice physical education and those who do not, the authors applied a 14-week intervention program and found significant differences in total QoL (*p* = 0.000), as well as in all QoL subscales, in children who practice physical education compared to those who do not. The same was observed in the study conducted by Sanpp et al. [45], where a program based on fundamental motor skills was applied in three different intervention period, in accordance with school breaks, with 10 consecutive weeks of intervention. After the intervention, the authors found an increase in emotional and social function in all intervention periods, an increase in physical function in the last two intervention periods and an increase in school function in the second intervention period.

Although most of the studies included in the systematic review found positive associations between PA, PE and sport and QoL, in the study carried out by Pérez et al. [44], the authors found no significant differences in any of the QoL domains assessed in the study, in the self-report version (self-esteem: *p* = 0.336; healthy habits: *p* = 0.187; leisure time: *p* = 0.220 and personal satisfaction: *p* = 0.987) or in the version applied to relatives/carers (*p* = 0.587), after a 12-week intervention in an aquatic environment. The authors suggest that the lack of improvement in terms of QoL may be related to the fact that the participants’ physical fitness did not display a significant increase after the intervention program [44]. The same was true in the study carried out by Fjellstrom et al. (2022), where there were no statistically significant differences in the QoL (*p* = 0.26; r = 0.006) of the individuals assessed after the web-based PA program, and it was even possible to observe a drop in the score for the “leisure activities” domain (*p* = 0.046). According to Fjellstrom et al. [41], this drop in scores can be explained by the fact that the study was carried out during the COVID-19 pandemic and there were various restrictions that may have affected the extent to which participants had the opportunity to enjoy leisure activities. In addition, in the post-intervention period, the restrictions were more severe than before the intervention [41], which may have influenced the QoL results.

### 4.2. Well-Being

With the aim of analyzing the relationship between sport and life satisfaction and understanding which type of sport is most associated with life satisfaction, Moltó and Bruna [17] found that there were no significant differences in life satisfaction (*p* = 0.37) between individuals who practice sport (28.36 ± 5.63) and those who do not (27.91 ± 4.60); however, participants who practice team sports are more satisfied with their lives (*p* = −2.06; r = 0.58) and value their living conditions more (*p* = −2.28; r = 0.59). Also, Bowers et al. [39], when analyzing the experiences and perspectives of athletes and their families in relation to the Special Olympics, found that for athletes, being part of the Special Olympics provides benefits for their physical health and well-being, represents a chance to socialize and establish relationships, and provides a sense of belonging and a sense of purpose. For family members, the Special Olympics essentially promotes family commitment and the creation of community social networks.

Barnet-Lopez et al. [38], when analyzing differences in the emotional well-being of adults with IDD after 26 dance sessions, found significant differences in the pre- and post-test scores of the experimental group (*p* = 0.007), while there were no differences in the control group (*p* = 0.560). The emotional indicators assessed were reduced in 13 participants in the experimental group.

In the study conducted by Carmeli and colleagues [6], after the PE intervention program, the authors found large differences in self-perceived well-being in the experimental group. The same happened in the study by Carmeli et al. [16], where the authors analyzed the effect of a PE program on the balance, strength and well-being of people with IDD. The two intervention groups used different PE programs, but both showed improvements in self-perceived well-being (group A: *p* = 0.05; group B: *p* = 0.05).

In contrast, in the study carried out by Shields et al. [46], in which participants performed 150 min of moderate to intense PA over 8 weeks, the results obtained are not in line with those mentioned above, as the authors found no significant differences between the control group and the experimental group ([95% CI]: −2.8 [−9.3, 3.6]) in terms of perceived well-being. According to the authors, the reason for these results is unclear, but they believe that it may reflect the physical effort that the participants had to exert to complete 150 min of at least moderate PA during the week [46].

Despite the diversity of objectives and methodologies, participation in PA, PE or sports programs seems to have positive and significant results in terms of QoL [6,15,18,40,42,43,45,47] and well-being [6,16,17,38,39] in people with IDD.

The results obtained in this systematic review seem to be corroborated by previous studies, such as the systematic review carried out by Bartlo and Klein [7], with the aim of systematically researching and critically examining the strength of research evidence on the effectiveness and feasibility of PA programs for adults with IDD, which verified the positive effects of PA on the perception of health and QoL in the population under study. The results are also corroborated by the systematic review carried out by Jacob et al. [32], in which the results obtained demonstrate the significant impact of PA on improving QoL in adults with IDD. Jacob et al. [32] aimed to identify the benefits of PA in adults with IDD and to analyze the possible impact of PA based on gender difference.

In addition to the aforementioned studies, others carried out with individuals with motor disabilities, veterans with different types of disabilities and the elderly demonstrate the positive effects of PA, PE and sport on QoL, such as the study by Ganesh and Mishra [48], in which the authors found a positive correlation between PA levels and all QoL domains (physical health: r = 0. 819, *p* < 0.050; psychological well-being: r = 0.776, *p* < 0.050; social relationships: r = 0.706, *p* < 0.050 and environment: r = 0.627, *p* < 0.050) in adults with motor disabilities. The study by Laferrier et al. [49] seems to confirm the positive relationship between the QoL perceived by the veterans themselves (r = 0.40, *p* < 0.001) and the number of years they have been practicing sport, physical exercise and/or recreational activities. The systematic review carried out by Wei et al. [31] also seems to corroborate these results, and found positive effects of PE on the QoL of healthy elderly people.

The same can be observed regarding well-being, with different studies corroborating the results obtained in this review, such as the systematic review by Windle et al. [50], in which the authors found a positive relationship between PE and PA and well-being in the elderly. Also, the results obtained in the systematic review by Borland et al. [51] seem to suggest a positive relationship between PA and psychological well-being in children and adolescents with IDD. The same can be seen in the study by Hassmen et al. [52], where the authors found that individuals with IDD who practiced PE more frequently had lower scores on questionnaires measuring negative affect and higher scores on measures of positive affect (e.g.,: depressive symptoms *p* < 0.001; perceived stress *p* < 0.01 and social integration *p* < 0.001), contributing to their psychological well-being [52].

### 4.3. Exercise Prescription

Taking into account the characteristics of the intervention programs of the studies included in this systematic review, the differences between them are prominent, with the duration of the interventions varying from eight weeks [46] to ten months [6], from one [47] to three sessions [41] a week and the duration of the sessions varying from 45 min [44] to 70 min [43]. However, it was possible to see two common points between most of the interventions—the warm up and a return to calm [15,16,38].

### 4.4. Limitations, Future Research and Practical Application

The authors of the articles included in this systematic review highlighted some limitations found in their studies that should be taken into account for future studies, such as the lack of validity and reliability of some of the tests applied [38,42,44,47], the selection and recruitment of participants that does not reflect the actual pattern of the population [42,46], the sample size that does not allow for the generalization of the results obtained [15,16,39,43,44,45,46,47], short intervention programs [15], the lack of a control group [44,45], interviewers not being blind to the grouping of participants [18], and PA being assessed subjectively rather than objectively [41].

It should also be noted that five of the studies identified had poor methodological quality, especially due to problems related to the low representativeness of the sample as well as the lack of calculation of the sampling power. Thus, as future recommendations, the authors of the studies emphasize the need to include people with severe IDD, not restricting the sample to people with mild and moderate IDD [40], to assess QoL using validated assessment scales [42,47], to use more meaningful and representative samples of the study population [15,16,17,38,39,40,43], to carry out subgroup analyses (e.g., by age, level of severity, gender) [17,38], to consider clinical and biological parameters and VO_2_ analysis [15], to analyze sports interventions [45], to analyze the impact of different sports on QoL and the impact of sport on different life cycles [17], and finally, to perform longitudinal and follow-up studies to identify the response to programs [15,18,39].

This systematic review addresses and shows a set of factors and benefits that support the integration of the practice of PA, PE or sport as a fundamental element in improving the QoL and well-being of people with IDD. It is important that the practice of these activities is carried out regularly, prescribed and monitored by professionals with basic training in sports science and specific training in IDD. Considering a multidisciplinary approach, integrated into primary and secondary health services, it is necessary to update intervention strategies for this population, including the incorporation of PA, PE or sport into their routines, which is fundamental for maintaining and improving physical fitness and functional capacity and, consequently, improving the QoL and well-being of people with IDD.

## 5. Conclusions

According to the studies included in this systematic review, we can conclude that the practice of PA, PE and sport seems to contribute to improving the QoL and well-being of people with IDD.

Despite the growing research interest in this area, there is still a notable lack of studies exploring the impact of these programs, especially sports-based programs, on the QoL and well-being of this population, with studies presenting small samples and unsatisfactory methodological quality. In light of this, more studies are needed to better analyze the characteristics of intervention programmers and their effects on the QoL and well-being of people with IDD. Specifically, it would be important to carry out RCT studies that analyze the role that practicing sports can have on the quality of life of people with different levels of severity of IDD, as well as analyzing the role of other variables in this relationship (e.g., level of education, socioeconomic status and employability).

## Figures and Tables

**Figure 1 healthcare-12-00654-f001:**
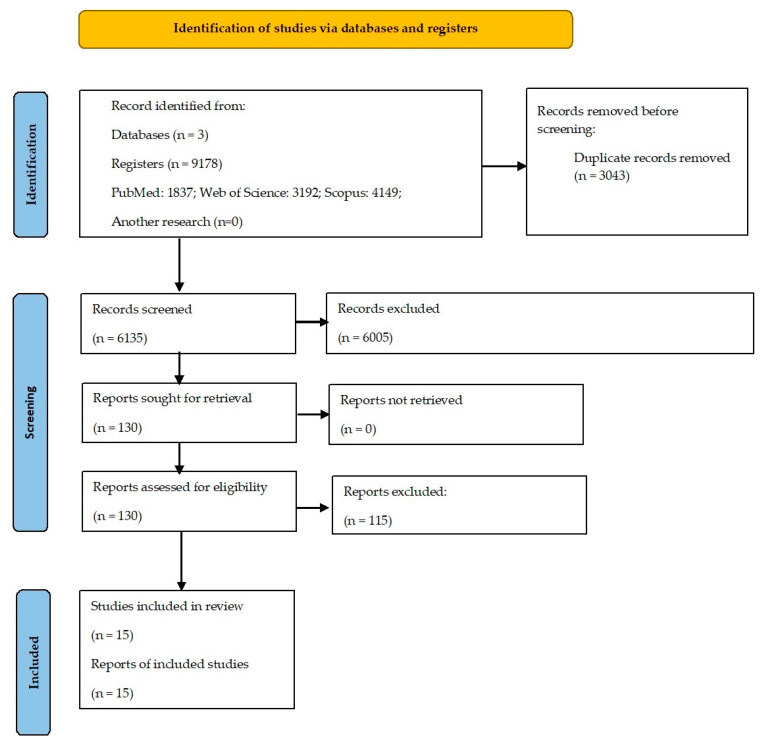
PRISMA flow diagram illustrating each phase of the search and selecting process.

**Table 1 healthcare-12-00654-t001:** Characteristics, results, and methodological quality of each of the studies included in the present review.

Studies	Aims	Participants	Duration/Frequency	Exercise/Intensity	Measurements	Results	Methodology Quality
Barnet-Lopez et al. [38], Spain, Europe	To analyze changes in the emotional well-being of adults with IDD after 26 sessions of dance/movement therapy.	N = 42 (male = 24; female = 18); 19–66 years old; moderate to severe IDD. Convenience sample. Randomized groups: EG (N = 22) and CG (N = 20). No sample size calculation.	3 months; 2× a week; 60 min/session.	Session phases: check-in, warm-up, transition-process and check out. Components and elements of the activities: body scheme, rhythms, self-concept, relationship, identification of the different types of emotions, Laban effort, balance and coordination, grounding and free dance.	Emotional well-being—Human Figure Drawing Test (HFD)	Experimental group (pre vs. post intervention): 5.55 ± 3.648 vs. 4.05 ± 3.258. Control group (pre vs. post intervention): 4.30 ± 0.669 vs. 4.50 ± 0.686. Significant differences between pre-test and post-test for the EG (*p* = 0.007) and no significant differences for the CG (*p* = 0.560). The emotional indicators are reduced in 13 participants from the EG, while in the CG, there were only 6 participants with reduced emotional indicators. Five emotional items increased their frequency after the program and fourteen emotional items decreased their frequency after the program, showing improvement.	Fair
Bowers et al. [39], Irland, Europe	To examine the experiences and perspectives of people with IDD, their families and staff who work with them about the Special Olympics on the island of Ireland.	N = 47 (no reference to sex); 15 athletes; 11 family members of athletes; 6 non-athletes; 7 family members of non-athletes and 8 staff members. No sample size calculation.	Not applicable	Not applicable	Focus group interviews; individual semi-structured telephone interviews; supplementary qualitative data extracted from four open ended questions contained in a quantitative survey used within the SOPHIE study.	For the athletes: benefits to physical health and well-being; the possibility of connecting socially; have a sense of purpose and belonging; they claim that the selection process is a barrier to progression. For athletes’ families: the Special Olympics promotes family commitment; community social networks; provided opportunities to witness their achievements and be proud; To promote participation in the Special Olympics: better information about the Special Olympics.	Poor
Carbó-Carreté et al. [40], Spain, Europe	To analyze the relationship between the practice of PA and the QoL of people with IDD.	N = 529 (male = 296; female = 233); 16-66 years old; borderline, mild, moderate and severe IDD. Convenience sample; no sample size calculation.	Not applicable	Not applicable	QoL—Personal Outcomes Scale–Spanish Adaptation;	The value obtained in the structural equation model (β11 = 0.703, *p* < 0.001) allows the authors to confirm that the levels of PA have an impact on the QoL of people with IDD. Thus, the data confirmed that PA acts as an important predictor of QoL improvement. The results present acceptable coefficients for the eight first-order factors (QoL domains) and for the three second-order factors (Independence, Social Participation and Well-Being). The well-being factor was the one that presented the highest values. A high value was also found in the Independence factor, specifically in the self-determination domain. The lowest value was associated with the Social Participation factor, in the domain of social inclusion, although this was considered a significant result.	Poor
Carmeli et al. [6], Israel, Asia	To understand whether there is a positive relationship between the perception of well-being and physical exercise among adults with IDD.	N = 60 (male = 14; female = 46); 46–77 years old. Mild IDD. Non-randomly selected sample; non-randomized groups: EG (N = 31) and CG (N = 29). No sample size calculation.	10 months; 3x a week; 40-45 min/session.	Warming-up movements; large body movements in sitting and standing for stability and flexibility; dynamic balance exercises; general strength training (light hand weights and elastic bands).	Self-perceived well-being—Harter’s Self-Perception Profile Modified; Health and QoL—Nottingham Health Profile (NHP).	Higher change in self-perception of well-being assessed by the NHP for the EG. After the intervention program, significant differences were found for the EG in the following NHP domains: energy (*p* = 0.001), social isolation (*p* = 0.001) and physical mobility (*p* = 0.001). The authors observed significant differences in the mean NHP questionnaire score between the initial assessment and the final assessment in the EG. Through an analysis of variance, the authors found a significant difference between the groups in the specific domains of social acceptance (F = 8.79) and physical appearance (F = 3.15).	Fair
Carmeli et al. [16], Israel, Asia	To analyze the effect of a physical exercise program on balance, strength and general well-being in adults with IDD.	N = 22 (male = 7; female = 15); 54–66 years old; mild IDD. Non-randomly selected sample, divided into two groups qualitatively and quantitatively: general group A (n = 10: 7 = female; 3 = male) and group B (n = 12; 8 = female; 4 = male). No sample size calculation.	6 months; 3× a week; 40–45 min/session.	Balance-training program (group A): warming up movements; dynamic balance exercises (i.e., toe-to-heel walk, tandem standing, side walking, dance, roll a ball, push, pull, catching and throwing). Muscle strengthening program (group A): in each session seven basic exercises were used (knee extension, knee flexion, ankle plantar flexion, hip extension, hip abduction, trunk flexion and trunk extension); 1–2 sets; 8–10 repetitions; resting period of 2–4 min between sets; 5–10 min to complete each muscle group. General exercise program (group B): warming-up movements; large body movements in sitting, standing and walking for general mobility, stability and flexibility.	Well-being questionnaire—the authors modified the Harter’s self-perception profile (Aasland and Diseth, 1999).	Both groups showed improvements in self-concept of well-being. Group A (pre training vs. post training): 68 ± 7 vs. 83 ± 9, *p* = 0.05. Group B (pre training vs. post training): 67 ± 8 vs. 85 ± 8, *p* = 0.05.	Fair
Crawford et al. [18], United Kingdom, Europe	To compare the impact of sports on the psychosocial behavior of people with IDD who (1) are Special Olympics athletes, (2) do not practice any type of sport and (3) practice physical exercise or sports, but not through the Special Olympics.	N = 101 (male = 57; female = 44); 18–67 years old. Special Olympics group n = 51 Mencap Sport group n = 20 Mencap No Sport group n = 30 Missing data for eight participants reduced the total sample for statistical analyses to 93. Power calculations were performed to establish appropriate sample sizes (calculations suggested a sample size of 40 in each group).	Not applicable	Not applicable	QoL—The Life Experiences Checklist (LEC);	The Special Olympics group reported differences between the groups on total scores of LEC (Special Olympics: 23.4 ± 3.7; Mencap Sport: 21.0 ± 3.5; Mencap no Sport: 21.2 ± 3.7; Group Comparison: x²(2, 101) = 10.1; *p* = 0.006), but also in the leisure and relationships subscales. No differences were found between the groups regarding the “opportunities” subscale.	Poor
Diz et al. [15]. Portugal, Europe	To analyze the effect of a regular PA program on the adaptive skills, motor competences, and QoL of institutionalized adults with IDD.	N = 16 (male = 7; female = 9); 24–61 years old; mild to moderate IDD. Convenience sample. Randomized groups: EG (N = 8) and CG (N = 8). No sample size calculation.	20 weeks; 2× a week; 50 min/session.	Active warm-up, core activities and stretching. PA program included rhythmic exercises, muscle strength and amplitude, spatial orientation, balance, body awareness, attention and memory.	QoL—Portuguese version of the Personal Outcomes Scale (P_POS)	In the last two assessment periods, better scores were recorded in most QoL domains for the EG, with a significant difference recorded between the pre-intervention and post-intervention periods, in the total quality of life score, in the proxy version (pre- vs. post-intervention: 97.75 ± 8.46 vs. 101.750 ± 7.69, *p* = 0.01, r = 0.32).	Fair
Fjellstrom et al. [41], Sweden, Europe	To explore the feasibility and effectiveness of a web-based PA training program for people with IDD. A secondary outcome was to assess enjoyment of the training program and QoL.	N = 22 (male = 12; female = 10); 18–60 years old; mild to moderate IDD. No sample size calculation.	12 weeks; 3× a week; 50 min/session.	150 min per week of moderate PA; Combination of strength exercises and resistance exercises length with balance and flexibility. Different progression levels were applied to be able to meet the participants requirements (e.g., the participants could choose between jumping and walking to be able to meet the moderate intensity level).	QoL—Manchester Short Assessment of Quality of Life (MANSA);	The mean scores of the satisfaction items were 5.9 ± 0.92 pre-training and 5.8 ± 0.9 post-training. No significant differences were observed in QoL (*p* = 0.26 and r = 0.006), except for “leisure activities”, where the score on the post-intervention tests was lower when compared to the score obtained before the intervention (*p* = 0.046).	Fair
Mercado et al. [42], Spain, Europe	To study the benefits of dance as a tool for improving the QoL of people with IDD.	N = 9 (male = 1; female = 8); 22–58 years old; 3 dance teachers, 1 mother, 1 president of an association and 4 persons with IDD. Convenience sample. Level of education: 45% primary, 33% secondary and 22% university studies.	Not applicable	Not applicable	Semi-structured interview: vision and perception of the group’s social inclusion and QoL.	The findings show the benefits and potential of dance regarding different dimensions of QoL and human functioning in IDD persons, like personal development, physical and emotional well-being and interpersonal relationships. The results indicate that dance benefits the QoL of people with IDD.	Poor
Moltó and Bruna [17], Spain, Europe	(1) To analyze the practice of sports as a significant activity for people with IDD; (2) to analyze the association of sport with life satisfaction and self-determination; (3) to establish which type of sport is most associated with life satisfaction and self-determination.	N = 74 (male = 49; female = 25); aged 18 or over; mild to moderate IDD. EG (N = 42) and CG (N = 32). No sample size calculation.	Not applicable	Not applicable	Sociodemographic and sports questionnaire (constructed ad hoc for this investigation); satisfaction with life—Satisfaction with Life Scale (SWLS); self-determination—Arc’s Self-Determination Scale.	The three main reasons why the participants participate in sport are as follows: (1) because they enjoy it and find it pleasurable, (2) for health reasons and (3) for the possibility of interacting with other people and making friends. No significant differences (*p* = 0.37) were found in SWLS between participants who played sports (28.36 ± 5.63) and those who did not (27.91 ± 4.60). Participants who practice team sports are more satisfied with their lives (*p* = −2.06; r = 0.58) and satisfied with the sporting activity (*p* = −3.63; r = 1.03). Also, individuals who practice team sports value their living conditions more (*p* = −2.28 and r = 0.59).	Poor
Ozkan and Kale [43], Turkey, Eurasian	To analyze whether there is a difference between the QoL and motor skills of children with IDD who participate in physical education activities for 14 weeks and those who do not.	N = 34 (male = 19; female = 15); 8–12 years old. EG 18 (male = 11; female = 7); CG: 16 (male = 8; female = 8). No sample size calculation.	14 weeks; 2× a week; 60–70 min/session.	Educational games; 1–8 weeks, basic skills were practiced. In weeks 9–14, the progress of children was considered and practices requiring more rapid movement and coordination were included.	QoL—Pediatric Quality of Life Inventory (PedsQL).	The inventory total score (*p* = 0.000) and all the other sub-scales of PedsQL of children with IDD who participated in the physical education activities program improved more, with significant differences, than children with IDD who did not participate in the program.	Fair
Pérez et al. [44], Spain, Europe	To analyze the potential benefits of an aquatic exercise program on the health-related physical fitness and QoL of a group of adults with DS and identify the impact that the program may have on the self-perception of QoL of parents or caregivers.	N = 14 (male = 7; female = 7); 21–49 years old. Moderate or severe IDD. No sample size calculation.	12 weeks; 2× a week; 45 min/session.	Warm up (15min): respiration exercises (time: 5 min; sets: 3 reps × 30 s × 2 sets; rest between repetitions: 5–10 s; rest between sets: 1 min; swimming speed: medium); crawl kicks while holding the edge of the pool (time: 5 min; sets: 3 reps × 30 s × 2 sets; rest between repetitions: 10 s; rest between sets: 1 min; swimming speed: medium-high); main part (30 min): crawl stroke (arms movement technique with pull buoy); crawl stroke (legs movement technique) and backstroke (legs movement technique)—time: 10 min; sets: 2 reps × 15 m × 3 sets; rest between repetitions: 10 s–passive rest; rest between sets: 1 min—active rest; swimming speed: high); cool down (5 min).	QoL—short modified version of a Spanish QoL questionnaire (applied to participants- self-report); QoL—World Health Organization Quality of Life-BREF (WHOQOL-BREF) (applied to parents/caregivers).	The authors did not observe significant changes, on the self-report scale, in any of the QoL dimensions assessed in the study, indicating that the exercise program did not have a significant impact on the QoL of the participants (self-esteem and health, test: 2.45 ± 0.74; retest: 2.59± 0.61, *p* = 0.336; healthy habits, test: 3.39 ± 0.56; retest: 3.14 ± 0.60, *p* = 0.187; leisure time, test: 2.90 ± 0.82; retest: 3.11 ± 0.57, *p* = 0.220; and personal satisfaction, test: 3.07 ± 0.54; retest: 3.07 ± 0.61, *p* = 0.987). The same happened with the perception of QoL of parents and caregivers in relation to the participants (total score, test: 13.53 ± 1.49; retest: 13.65 ± 1.384, *p* = 0.587).	Fair
Snapp et al. [45], USA, North America	To provide evidence regarding the positive impact of interventions based on fundamental motor skills on the QoL of children with IDD.	N = 10 (no reference to sex); 4–14 years old; ASD, speech deficits, DS and fragile syndrome. No sample size calculation.	Three separate 10- week interventions; 1× a week; 1 h/session.	Each session followed the same general routine: 10–15 min of free play, 5 min warm up, 35 min of direct instruction, and 5 min cool down.	The instrument was applied six times, at the beginning and end of each 10-week intervention; QoL—Pediatric Quality of Life Inventory 4.0 (PedsQL);	Emotional function: an increase during the three interventions was observed, with a slight increase between the end of the first and the beginning of the second intervention and a decrease between the end of the second and the beginning of the third intervention. Social function: an increase during the three interventions was observed, with a continuous increase between the end of the first and the beginning of the second intervention and a decrease between interventions two and three. Physical function: a decrease during the first intervention and an increase during the second and third interventions were observed. School function: an increase was noticeable during the second intervention.	Fair
Shields et al. [46], Australia, Oceania	To investigate whether a PA program designed according to the Rimmer and Rowland (2008) framework was viable, whether the program was safe, and whether it led to improvements in walking ability, and to assess risk factors associated with chronic diseases and positive changes in participants’ perceptions of well-being and changes in PA.	N = 16 (male = 8; female = 8); 18–35 years old; mild to moderate IDD. Convenience sample. Randomized groups: EG (N = 8) and CG (N = 8). No sample size calculation.	8 weeks;	150 min of moderate-intensity PA per week—two 45 min walks per week with a student mentor and another 60 min of PA (independent—without mentor)	Perceptions of well-being—The Life Satisfaction Scale.	There were no significant differences between the groups for perceptions of well-being. Intervention group (pre vs. post): 29.8 ± 7.5 vs. 28.2 ± 7.5. Control group (pre vs. post): 30.6 ± 4.6 vs. 31.3 ± 4.8. Difference between groups (Week 9–Week 0): −2.8 [−9.3, 3.6].	Good
Tomaszewski et al. [47], USA, North America	(1) To describe the average PA for adults with ASD and IDD using PA meters, (2) to describe the QoL of adults with ASD and IDD and (3) to examine the relationship between PA counts steps and QoL.	N = 38 (71.1% male); 18–55 years old. Non-convenience sample. No sample size calculation.	All waking hours for 1 week.	Not applicable	QoL—Quality of Life Questionnaire (QOL-Q)	Satisfaction domain was significantly higher (23.37 ± 2.721) than the domains of competence (18.89 ± 6.75), independence (20.61 ± 3.45) and social belonging (21.28 ± 3.16). The competence domain and the total QoL value were significantly associated with weekly step count (*p* = 0.004, r = 0.46 and *p* < 0.001, r = 0.56). The total QoL score was regressed onto average steps per day, nonverbal IQ and age. Average steps per day, nonverbal IQ, and age accounted for 34.2% of the variance in quality of life. For every 1000-step increase in steps per day, the total QoL score increases by 2.56 points.	Poor

Note: CG, control group; EG, exercise group; min, minutes; IDD, Intellectual and Developmental Disability; N, participants; PA, physical activity; QoL, quality of life.

## Data Availability

Data are contained within the article.

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
