# Peer review of "Physical Activity, Quality of Live and Well-Being in Individuals with Intellectual and Developmental Disability"

_healthcare, 2024, doi:10.3390/healthcare12060654_

Round 1

Reviewer 1 Report

Comments and Suggestions for Authors

The authors reviewed systematically the papers to assess the state of the art on the role of physical activity, exercise and sport in the quality of life and well-being of people with intellectual and developmental disability, seeking to understand the current panorama in this area and provide answers to these questions.

The study entitled "Physical activity, quality of live and wellbeing in individuals with Intellectual and Developmental Disability" has the potentiality of being shared with the scientific community, but it would benefit from a minor revision.

Develop these important parts:

Introduction: The introduction is scarce and although it names the variables, it does not indicate whether there are studies that have related them either in the same study population or in others. Likewise, there is no thorough review of the state of the art to justify the need for this work.
The authors should clearly describe the scientific evidence that supports the hypothesis.

Abstract: start with a brief background and include purpose, methods, results, and conclusions of the report.

Regarding the methodology, the authors indicate that they use the prism methodology, although they should detail it more concretely, mainly in the process of filtering the results and relating it to the inclusion and exclusion criteria.

Search strategy: Specify the methods used to assess risk of bias in the included studies (with graph)

Discussion: In the discussion, the authors should emphasize the importance of their findings for the study population and highlight future lines of work in a more extensive and detailed manner. The discussion and introduction demonstrate that the review of papers could have been more in-depth and even encompassed more studies. Provide a deeper interpretation of the results in the context of other evidence.

I suggest adding a practical application section with the most important recommendations.

Author Response

Response to REVIEWER 1

The authors reviewed systematically the papers to assess the state of the art on the role of physical activity, exercise and sport in the quality of life and well-being of people with intellectual and developmental disability, seeking to understand the current panorama in this area and provide answers to these questions.

The study entitled "Physical activity, quality of live and wellbeing in individuals with Intellectual and Developmental Disability" has the potentiality of being shared with the scientific community, but it would benefit from a minor revision.

Develop these important parts:

Introduction: The introduction is scarce and although it names the variables, it does not indicate whether there are studies that have related them either in the same study population or in others. Likewise, there is no thorough review of the state of the art to justify the need for this work.

Response: Thanks for reviewer, we added information on introduction.

The authors should clearly describe the scientific evidence that supports the hypothesis.

Response: Thanks for reviewer, we added information on introduction.

Abstract: start with a brief background and include purpose, methods, results, and conclusions of the report.

Response: Dear reviewer, the abstract has been revised.

Regarding the methodology, the authors indicate that they use the prism methodology, although they should detail it more concretely, mainly in the process of filtering the results and relating it to the inclusion and exclusion criteria.

Response: We revised the 2.3. Selection and data collection process.

Search strategy: Specify the methods used to assess risk of bias in the included studies (with graph)

Response: Dear reviewer, considering the various types of studies that were extracted, we did not analyze the risk of bias of the studies. We only analyzed their methodological quality.

Discussion: In the discussion, the authors should emphasize the importance of their findings for the study population and highlight future lines of work in a more extensive and detailed manner. The discussion and introduction demonstrate that the review of papers could have been more in-depth and even encompassed more studies. Provide a deeper interpretation of the results in the context of other evidence.

Response: Dear reviewer, considering our eligibility criteria, only 15 studies met them. These criteria were defined to respond to a clear study aims. In any case, we have reformulated our introduction and discussion.

I suggest adding a practical application section with the most important recommendations.

Response: We introduce a practical application section.

Reviewer 2 Report

Comments and Suggestions for Authors

I do not think that this paper adds much to existing knowledge. It is a literature review, limited to few articles which are analyzed in depth (only 15), some of them assessed as "poor" according to its methodological quality. And the only conclusions of the paper are that "PE and sport seem to contribute to improving the QoL and well-being of 356 people with IDD" (lines 356-357), and that more and better studies are needed on this matter.

I think that this paper should be deeply revised.

On the one hand, I suggest selecting a larger number of articles for complete analysis: the authors tell us that 130 potentially relevant articles were identified, and, after reading the articles in full, a sample of only 15 articles was considered for complete analysis. I suggest that a more flexible application of the eligibility and exclusion criteria would probably provide a bigger sample of articles which deserve to be analyzed, what would increase the interest and relevance of the paper. I would also suggest to consider papers on the relation between physical activity and employability of persons with IDD, because employability increases quality of life and well-being, and therefore a positive correlation between physical activity and employability suggests also indirectly a positive correlation between physical activity and Qol/well-being.

Second. In the discussion of the articles, I think a section should be included on methodological quality of the papers which have been reviewed in depth. It would be interesting for the reader to know which is the average methodological quality of the papers, which are the most frequent weaknesses or flaws, etc. If this literature review is aimed to provide some orientations for future research, it would be interesting to know how to improve the methodological quality of the studies on the matter, specially if some of them have been evaluated as poor.

Third. I think that the conclusions of the paper should be more developed. It is not enough to say that more papers on this matter are needed, the authors should also suggest with more detail and concretion possible orientations on methodology and content of future research, based on the papers which have been analyzed and discussed.

Author Response

Response to REVIEWER 2

I do not think that this paper adds much to existing knowledge. It is a literature review, limited to few articles which are analyzed in depth (only 15), some of them assessed as "poor" according to its methodological quality. And the only conclusions of the paper are that "PE and sport seem to contribute to improving the QoL and well-being of 356 people with IDD" (lines 356-357), and that more and better studies are needed on this matter.

Response: Thank you for your comments. We recognize that the number of articles is a limitation of this work, but on the other hand it is evidence of the relevance and urgency of studying this topic and studying with better methodologic quality (e.g., lack of calculation of the sampling power; lower number of participants and representativity). The discussion has also been improved in an attempt to highlight the main findings of the work and the practical implications resulting from this study.

I think that this paper should be deeply revised.

Response: We will try to respond to the reviewer's comments and improve the manuscript with the indications and suggestions made.

On the one hand, I suggest selecting a larger number of articles for complete analysis: the authors tell us that 130 potentially relevant articles were identified, and, after reading the articles in full, a sample of only 15 articles was considered for complete analysis. I suggest that a more flexible application of the eligibility and exclusion criteria would probably provide a bigger sample of articles which deserve to be analyzed, what would increase the interest and relevance of the paper. I would also suggest to consider papers on the relation between physical activity and employability of persons with IDD, because employability increases quality of life and well-being, and therefore a positive correlation between physical activity and employability suggests also indirectly a positive correlation between physical activity and Qol/well-being.

Response: We thank the reviewer comments and suggestions. However, the eligibility criteria were defined on the basis of what we considered appropriate for the purpose of this work, which sought to analyze the state of the art on the role of physical activity, exercise and sport in the quality of life and well-being of people with intellectual and developmental disability.

We also recognize the importance of including other variables in future studies (e.g., employability). However, this was not the aim of this study. Thus, the suggestions for future studies were improved by including some of these aspects.

Second. In the discussion of the articles, I think a section should be included on methodological quality of the papers which have been reviewed in depth. It would be interesting for the reader to know which is the average methodological quality of the papers, which are the most frequent weaknesses or flaws, etc. If this literature review is aimed to provide some orientations for future research, it would be interesting to know how to improve the methodological quality of the studies on the matter, specially if some of them have been evaluated as poor.

Response: The discussion has been improved based on the reviewer's suggestion.

Third. I think that the conclusions of the paper should be more developed. It is not enough to say that more papers on this matter are needed, the authors should also suggest with more detail and concretion possible orientations on methodology and content of future research, based on the papers which have been analyzed and discussed.

Response: The conclusion has also been improved with a clearer and more objective indication of possible methodological guidelines to consider.

Round 2

Reviewer 1 Report

Comments and Suggestions for Authors

The authors fail to address their manuscript according to reviewer suggestion. The manuscrip presents a number of issues that would need a general reinterpretation.

Author Response

Response to REVIEWER 1

The authors fail to address their manuscript according to reviewer suggestion. The manuscrip presents a number of issues that would need a general reinterpretation.

Response: Dear reviewer, thank you very much for your comments. However, we don't understand your review since, during the first round of revisions, we considered all your comments and suggestions which were carefully inserted into the manuscript.

We think there may be an error with the file we submitted or that the reviewer downloaded. We have therefore attached all the answers we gave in the first round, this time in greater detail.

We have also attached the revised manuscript, with the changes noted.

However, if the reviewer feels that any of the answers are insufficient, we are happy to improve them.

Reviewer's comments in the first review round

- Introduction: The introduction is scarce and although it names the variables, it does not indicate whether there are studies that have related them either in the same study population or in others. Likewise, there is no thorough review of the state of the art to justify the need for this work.

The authors should clearly describe the scientific evidence that supports the hypothesis.

Response: Thanks for reviewer, we added information on introduction (lines 65-73): “Nowadays, maintaining or improving QoL and well-being has been seen as a uni-versal goal throughout the life of individuals with IDD, requiring the identification or development of facilitating tools and strategies [1]. Although several systematics re-views show that the practice of PA, PE or sport improves these variables in the adults and elderly population without disabilities [e.g., 26,27], studies in the population with IDD are scarce and inconclusive. The study by Carmeli et al. [33], examining anxiety and QoL assessments revealed a 50% enhancement in the exercise intervention group, a 38% amelioration in the leisure activity group, and no improvement in the control group, providing indication that these variables may be related.”

Abstract: start with a brief background and include purpose, methods, results, and conclusions of the report.

Response: Dear reviewer, the abstract has been revised (lines 10-12): “The practice of physical activity, exercise and sport has many benefits for the general population, but studies on the population with intellectual and developmental disabilities (IDD) are scarce and inconclusive.”

Regarding the methodology, the authors indicate that they use the prism methodology, although they should detail it more concretely, mainly in the process of filtering the results and relating it to the inclusion and exclusion criteria.

Response: We revised the 2.3. Selection and data collection process (lines 100-112): “The eligibility criteria have defined by all authors. The selected studies were re-viewed in their entirety by two independent reviewers (SD and MJ), considering the eligibility criteria’s.”

Search strategy: Specify the methods used to assess risk of bias in the included studies (with graph)

Response: Dear reviewer, considering the various types of studies that were extracted, we did not analyze the risk of bias of the studies. We only analyzed their methodological quality.

Discussion: In the discussion, the authors should emphasize the importance of their findings for the study population and highlight future lines of work in a more extensive and detailed manner. The discussion and introduction demonstrate that the review of papers could have been more in-depth and even encompassed more studies. Provide a deeper interpretation of the results in the context of other evidence.

Response: Dear reviewer, considering our eligibility criteria, only 15 studies met them. These criteria were defined to respond to a clear study aims. In any case, we have reformulated our introduction and discussion. Regarding future lines of research and practical implications, information has been added to the report (please see reply to the following comment).

I suggest adding a practical application section with the most important recommendations.

Response: We introduce a practical application section (lines 368-376): “This systematic review addresses and show a set of factors and benefits that support the integration of the practice of PA, PE or sport as a fundamental element in im-proving the QoL and well-being of people with IDD. It is important that its practice is carried out regularly, prescribed and monitored by professionals with basic training in sports science and specific training in IDD. Considering a multidisciplinary approach, integrated into primary and secondary health services, it is necessary to update inter-vention strategies for this population, including the incorporation of PA, PE or sport into their routines, which is fundamental for maintaining and improving physical fit-ness, functional capacity and, consequently, for improving the QoL and well-being of people with IDD.”

Reviewer 2 Report

Comments and Suggestions for Authors

I still find that the number of articles that have been analyzed in depth in this literature review is a little bit scarce, but, after reading the response of the authors of the article to my report and the improvements that have been made in the new version, I think that the paper is suitable for publication.

Author Response

Response to REVIEWER 2

I still find that the number of articles that have been analyzed in depth in this literature review is a little bit scarce, but, after reading the response of the authors of the article to my report and the improvements that have been made in the new version, I think that the paper is suitable for publication.

Response: Thank you very much for your positive feedback. We have tried to improve the manuscript considering the reviewers' recommendations.
